# TransES-ETA: A Novel Transformer-based Explainable and Efficient Structure for Predicting Estimated Time of Vessel Arrival

## Abstract

The estimated time of arrival (ETA) prediction is crucial in maritime AI for improving maritime shipping operation efficiency and resilience; however, they currently still face significant challenges. Traditional machine learning-based methods struggle with extracting accurate representations of Automatic Identification System (AIS) data, while neural network-based methods rely heavily on data quality and lack of explainability. In this paper, we propose TransES-ETA, a transformer-based, multi-task framework that jointly handles port call scheduling and ETA prediction in an end-to-end and explainable manner, leveraging the strong correlation between the two tasks. We first design a novel AIS data extractor composed of multiple modules, each responsible for processing different semantic data attribute categories, enabling the transformer to capture accurate representations. Then, the port call schedule module of our proposed pipeline predicts vessel trajectories and generates features that serve as inputs for the subsequent ETA prediction module. Finally, the ETA module aggregates both AIS-derived features and port call schedule features to produce the final ETA estimate. The entire multi-task pipeline is trained in an end-to-end manner, allowing for the simultaneous optimization of both tasks with the shared feature and thereby improving training efficiency. Furthermore, by incorporating port scheduling information, the ETA predictions become more interpretable. Comprehensive evaluations demonstrate that our proposed model achieves mean absolute error (MAE) 2.46h in ETA prediction. Additionally, the pipeline accelerates training and inference by 80.93% and 8.84% respectively, making the framework more efficient.

## 1 Introduction

The estimated Time of Arrival (ETA) prediction for each vessel is critical to enhancing the efficiency, reliability, and sustainability of maritime logistics. Accurate ETA predictions enable optimized vessel speeds, reducing fuel consumption and emissions, while supporting just-in-time scheduling that minimizes congestion and improves berth and resource allocation. By integrating ETA prediction with port call scheduling, shipping lines and terminals can streamline operations, reduce costs, and enhance supply chain reliability. These capabilities also form a foundation for smart ports, predictive traffic management, and autonomous maritime systems, making research into advanced, data-driven methods—leveraging Automatic Identification System (AIS) data and port activity—essential for building an adaptive and intelligent maritime ecosystem.

Traditional machine learning-based ETA prediction methods often rely on two distinct models: one for route prediction and another for time estimation. This decoupled structure prevents the extraction of a shared representation from AIS data, resulting in inconsistent data representations across tasks. Consequently, the learned representations tend to be closely tied to specific tasks as well as particular geographical regions or vessel routes, making the predictions often unstable and fail to generalize effectively to other datasets. Moreover, the sequential nature of these models causes errors from the route prediction stage to propagate and compound in the time estimation stage, further degrading the overall accuracy of ETA predictions (Xu et al. (2021); Ogura et al. (2021)).

Current data-driven approaches, including neural networks or transformer architectures for ETA prediction (Shipley et al. (2022)), struggle with poor data quality and the limited interpretability of end-to-end pipelines. To address them, we introduce TransES-ETA, a transformer-based multi-task framework to jointly predict port call schedules and ETAs. The architecture adopts an end-to-end training framework, eliminating the need for sequential, task-specific models. This design not only contributes to computational efficiency during both training and inference but also mitigates error propagation across separate model stages. First, a shared Transformer Encoder of this framework captures temporal and spatial dependencies within AIS data via a semantic data feature extraction module and generates a unified feature representation. Then, these shared representations are subsequently utilized by task-specific decoder branches, enabling cross-task information sharing and enhancing the robustness of AIS feature. This design not only mitigates error propagation commonly observed in two-stage models but also delivers superior generalization and predictive accuracy compared to single-task baselines.

In addition to accurate representation learning and efficient training and inference, the multi-task pipeline—along with the shared feature design—addresses the critical need for structural explainability in ETA prediction. Conventional deep learning–based end-to-end pipelines predict ETAs directly from raw AIS inputs by learning historical data patterns, but they often overlook the semantic and operational factors that are critical to accurate and explainable ETA prediction. Our framework incorporates port call schedule prediction as a pre-task, which plays a crucial role in guiding and improving the accuracy of the final ETA prediction. Instead of predicting only the final ETA, our model produces interpretable intermediate outputs—namely, the calling ports—which explain how the final estimate is derived and enhance the overall transparency and reliability of the prediction.

The encoding technique proposed by Kim et al. (2023), which reshapes international vessel trajectories between ports into a nested sequence structure, can effectively extract spatiotemporal features from AIS data, and this method is adopted in this paper. However, a GRU-based module in this method cannot batch process the data in this study, which reduces the training speed of the model. Therefore, in the semantic data feature extraction module, this paper proposes a novel graph-based method to replace the GRU-based approach used in previous studies, as an important improvement strategy. The model based on this strategy can process multiple sets of data in parallel, significantly accelerating the training and inference of ETA prediction. Experimental results show that compared with previous studies (Kim et al. (2023)), it can accelerate the training and inference stages by 80.93% and 8.84% respectively, greatly improving the operational efficiency of the model.

## 2 RELATED WORK

### 2.1 TRANSFORMER-BASED TRANSPORT DATA ANALYTICS

Transformer-based models, originally developed for natural language processing (Vaswani et al. (2017)), have become central to transportation analytics by leveraging self-attention to capture long-range temporal and spatial correlations. In the transportation domain, transformers have been successfully applied to diverse tasks, including traffic flow forecasting (Zeng et al. (2023); Li et al. (2023)), travel time estimation (Wu et al. (2021)), trajectory and route prediction (Jin et al. (2023); Kim et al. (2023)), and vessel or vehicle ETA prediction (Tang et al. (2023); Liu et al. (2024)).

These models excel in integrating heterogeneous and sequential inputs such as GPS and AIS streams, traffic sensor data, weather conditions, and contextual variables. Variants such as Temporal Fusion Transformers (TFT) (Lim et al. (2021)) and Spatial-Temporal Transformers (Xu et al. (2020); Zhang et al. (2023)) further enhance performance by incorporating attention-based temporal encoding and spatial graph structures, often in combination with Graph Neural Networks (GNNs) or convolutional modules to model transportation networks.

### 2.2 ETA PREDICTION

Traditional machine learning approaches for vessel ETA prediction have primarily relied on statistical regression models, tree-based algorithms such as decision trees and random forests, and ensemble techniques like gradient boosting (Shipley et al. (2022); Xu et al. (2021); Ogura et al. (2021)). These methods typically frame ETA prediction as a supervised regression problem, using handcrafted features derived from AIS data, historical voyage records, weather conditions, and port

Table 1: Overview of the Input Data

| Data | Description |
|------|-------------|
| *Grid Area* | $g_{1:T} = \{g_1, \ldots, g_N\} \in \mathbb{R}^{N \times (\lambda, \phi)}$ |
| *Local Pattern* | $\{x_{1:m_1}^1, \ldots, x_{1:m_N}^N\} \in \mathbb{R}^{N \times M \times f}$ for $M = \{m_1, \ldots, m_N\}$ |
| *Time Distance* | $\{\Delta_1, \ldots, \Delta_N\} \in \mathbb{R}^N$ |
| *Departure Port* | $P_x \in \mathcal{P}$ |
| *Ship Type* | $S_x \in \mathcal{S}$ |

activity. While such models are computationally efficient and relatively interpretable, they face several inherent limitations. First, their reliance on static or pre-engineered features restricts their ability to capture the complex temporal and spatial dependencies inherent in vessel trajectories. Second, they are highly sensitive to data sparsity, noise, and missing values in AIS streams, which can significantly degrade predictive accuracy Nguyen et al. (2020). Finally, these models lack adaptability to dynamic maritime environments, such as sudden weather changes, congestion, or route deviations, leading to suboptimal performance compared to modern deep learning or hybrid approaches that leverage sequential modeling and multimodal data fusion Yang et al. (2022); Chen et al. (2023).

Research based on deep learning models typically models input data as sequential data and treats ETA prediction as an end-to-end supervised task. Experiments have shown that their performance is superior to that of machine learning models (El Mekkaoui et al. (2023); Kim et al. (2023)). Although these models achieve high prediction accuracy, they have low interpretability. Such interpretability is particularly important in decision-making processes for applications such as traffic management, as stakeholders require transparency regarding model inputs. Therefore, researchers have been committed to improving the interpretability of deep learning-based frameworks.

## 3 SEMANTIC DATA FEATURE EXTRACTION MODULE

### 3.1 SEMANTIC DATA TRANSFORMATION

The purpose of data transformation operations is to adjust the data into a form suitable for the TransES-ETA. AIS data often has inaccuracies and irregular time intervals. Following Kim et al. (2023), who reconstruct trajectories into nested sequences to reduce spatiotemporal biases, we adopt and enhance this approach in our data encoding. The specific approach is as follows: First, denote the untransformed AIS sequence data at time step $T$ as $x_{1:T} = \{x_1, \ldots, x_T\} \in \mathbb{R}^{T \times f}$, where $f$ refers to the number of AIS information elements at a given timestamp, including speed, position, heading, etc. Rasterize this AIS sequence data into $\{G_1, \ldots, G_N\}$, where each grid element $G_k = \{g_k; x_{1:m_k}^k\}$ for $k \in \{1, \ldots, N\}$ represents a center coordinate $(\lambda, \phi)$ tuples of the spatial grid (denoted as $g_k$), and a subset of the aforementioned AIS sequence data (denoted as $x_{1:m_k}^k = \{x_{j+1}^k, \ldots, x_{j+m_k}^k\} \subset x_{1:T}$), where $m_k$ is the number of AIS message in a grid, $N$ is the number of grids, $j = \sum_{i=1}^{k-1} m_i$ and $T = \sum_{i=1}^{N} m_i$. In addition to this, we add the information of time distance $\Delta_{1:T} = \{\Delta_1, \ldots, \Delta_T\} \in \mathbb{R}^T$, which is the time intervals between each last observation in grids and the very first from the trajectory, denoted as $\{\Delta_1, \ldots, \Delta_N\} \in \mathbb{R}^N$, where $\Delta_N = \text{time}_\Delta(x_{m_N}^N - x_1)$ and the intervals are measured in hours, departure port $P_x \in \mathcal{P}$  ($\mathcal{P}$ is a set of ports), and ship type $S_x \in \mathcal{S}$  ($\mathcal{S}$ is a set of ship types). All inputs of the model are denoted as $\{G_{1:N}; \Delta_{1:N}; P_x; S_x\}$, as shown in Table 1.

We visualized the above information. In the sub-figure (a) of Fig. 1, black boxes represent grids, the sequence of blue dots within each box denotes the rasterized AIS data sequence, the orange dot inside the grid indicates the grid center, and the time difference between the last blue dot in the grid and the first blue dot of the entire trajectory is the time distance. Red dots represent departure ports, green dots represent arrival ports, and pink dots represent calling ports.

From a port operator's perspective, ETA prediction requires frequent updates, making it a dynamic regression task. Using a sliding window to sample five data types allows each sample to output an updated ETA. Noted, the details of AIS data are shown in the supplementary materials.

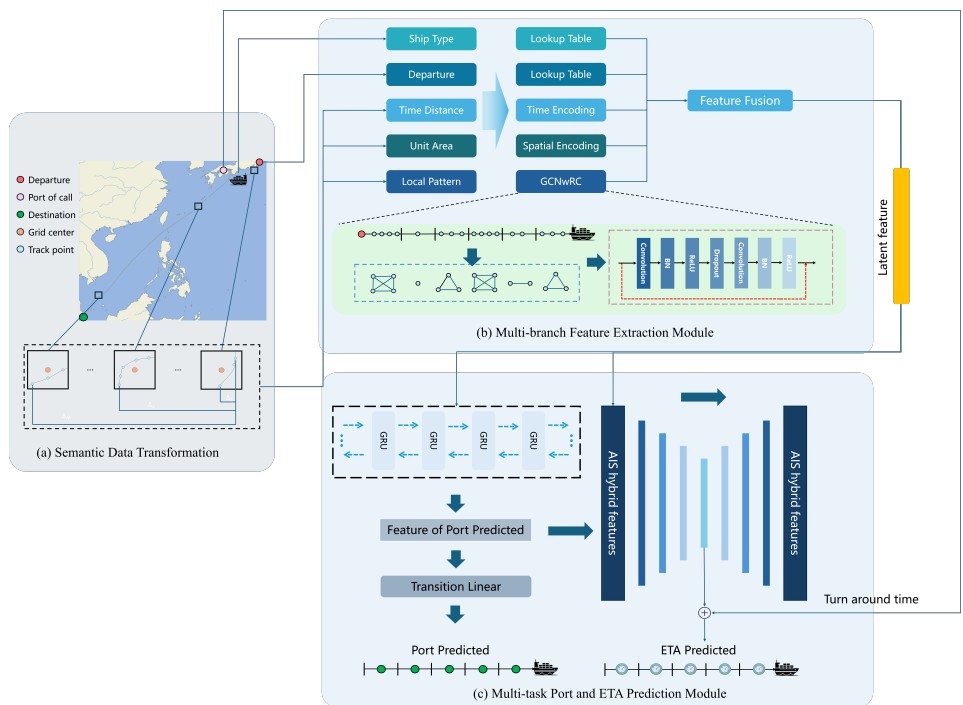

Figure 1: Visualization of the Overview Structure.

## 3.2 MULTI-BRANCH FEATURE EXTRACTION

We encode the above five types of information using formulas and deep learning models. The spatial encoding (SE) module aims to convert real-valued coordinates into high-dimensional vectors. In this study, the position encoding formula in Transformer Vaswani et al. (2023) is used to encode the *Grid Area*. That is, based on the spatial encoding formula in the appendix, $x \in \mathbb{R}^{N \times (\lambda, \phi)}$ is encoded into $x \in \mathbb{R}^{N \times d}$.

In the original text of this encoding method Kim et al. (2023), Gated Recurrent Unit (GRU) was used to encode *Local Pattern* information. Because of the input of GRU $x \in \mathbb{R}^{B \times L \times D}$ ($B$ is batch size, $L$ is sequence length and $D$ is feature dimension), the authors encoded $x^k \in \mathbb{R}^{m_k \times f}$ in batches over $N$ sequence steps. This is effective in the task of the original text, but it cannot encode the information of this article. As mentioned above, this paper uses a sliding window to dynamically sample data, obtaining data $x \in \mathbb{R}^{B \times N \times M \times f}$. Obviously, if GRU is to be used to process the data in this paper, batch processing will not be possible. The inability to batch process will increase the model training time, thereby raising the time cost of the experiment.

In view of this situation, we propose a Graph Convolutional Network with residual connections (GCNwRC) module to encode *Local Pattern*. As shown in the sub-figure (b) of Fig. 1, first, the AIS sequence data in each grid is encoded into a directed (bidirectional) complete graph. That is, each AIS message in the grid serves as a node, and all nodes are connected to each other. A graph composed of all grids within a time window forms an overall graph, representing *Local Patterns* within this time window. Since there are no connections between individual graphs, information from different grids will not be fused when encoding features using GCNwRC. This ensures data independence and avoids information loss or distortion caused by data fusion. A time window of sequence data contains $N$ graphs, with $m_k$ nodes in each grid, and each node has $f$ dimension feature. GCNwRC consists of two convolutional layers (details are provided in the appendix), two batch normalization layers, two ReLU layers, and one dropout layer. The red dashed lines represent residual connections, linking the start of the module to the end of the module. These connections are used to improve model performance and are inspired by the residual network mechanism.

The *Departure Port* and *Ship Type* are both semantic information. One of the most effective ways to encode semantic information is embedding in deep learning, which maps discrete inputs (such as words) into a continuous vector space. We assign $d$-dimensional vectors to each of departure

Table 2: Overview of the Dataset

| Indicator | Description |
|---|---|
| Ship Type | Crude Oil Ship and Container Ship |
| Covered sea areas | The South China Sea, the East China Sea, the Yellow Sea, and other maritime areas |
| The total number of departure ports and calling ports | 51 ports in countries like China and South Korea |
| Destination port | Singapore port |
| The average voyage distance | 2467.94 nautical miles |
| The total duration of all voyages | 45034.35 hours |

ports and ship types, such that the $d$-dimensional vectors corresponding to the indices of a specific departure port and ship type can describe the information of that port and ship type.

In the **Semantic Data Transformation** section, we convert timestamps into ship navigation time intervals, which represent the time distance between the last AIS message in a grid and the first observation of the entire trajectory. These intervals consist of a series of irregular time gaps. Following the approach in (Kim et al. (2023)), we obtain a time encoding formula (details are provided in the appendix) by modifying the Transformer positional encoding formula for *Time Distance*.

The above four encoding methods encode the corresponding information into $x \in \mathbb{R}^{N \times d}$. The output of the temporal encoding is added to the two types of data encoded by embedding, ultimately forming feature $x \in \mathbb{R}^{4 \times N \times d}$.

## 4 MULTI-TASK PORT AND ETA PREDICTION

To enhance the robustness and interpretability of the model, a Transformer-based multi-task prediction module is proposed, which is divided into the port prediction branch and the ETA prediction branch, as shown in sub-figure (c) of Fig.1.

### 4.1 PORT CALL PREDICTION MODULE

In the **Semantic Data Transformation** section, when processing data, the ship's departure port, calling ports, and destination port will be recorded, which is the ship's port call schedule. However, we do not explicitly use these features directly as inputs to the model. Instead, we infer the ship's next arrival port (which could be either a calling port or a destination port) based on the four-channel features obtained from **Data Feature Extraction**. This approach can not only enhance the interpretability of ETA prediction by predicting the ship's route information but also obtain rich features from another perspective of AIS sequence data through the feature extractor of the port prediction module, thereby further improving the performance of the ETA prediction module.

The above four-channel features are sequential in nature. After average pooling, a Bi-GRU model can be used to extract their features. By processing the sequence with two GRU units (forward and backward), the Bi-GRU captures contextual relationships in both directions, enabling it to model long-range bidirectional dependencies. As a lightweight alternative to Bi-LSTM, the Bi-GRU has fewer parameters, which reduces computational costs while preserving performance. Consequently, this study adopts the Bi-GRU as the feature extractor for port prediction.

In the multi-task architecture designed in this paper, the feature vectors output by Bi-GRU are decomposed into two parallel processing paths. The first feature branch is directly connected to the fully connected layer. Through layer-by-layer nonlinear transformation, this module maps the features extracted by Bi-GRU to the classification decision space, achieving accurate classification of the input temporal data. The second feature branch, as AIS sequence features from a different perspective, is integrated into the ETA prediction module, and the details of this part will be elaborated in the next section.

### 4.2 ETA PREDICTION MODULE

The AIS sequence features processed by **Data Feature Extraction** and **Port Call Prediction Module** exhibit both long-term dynamic variation characteristics and complex spatial association in-

formation. Traditional encoders, such as Recurrent Neural Networks (RNNs) and their variants LSTM and GRU, although capable of capturing sequential dependencies, are still inadequate for ETA prediction due to its high complexity. Convolutional Neural Networks (CNNs), while adept at extracting local features, struggle to establish long-range sequential associations. In contrast, the Transformer Encoder, leveraging its self-attention mechanism, can accurately capture long-range dependencies between any time steps in the sequence through global parallel computation, without being constrained by sequence length. Simultaneously, the multi-head attention mechanism can extract features from multiple dimensions in parallel, enabling in-depth mining and fusion of local spatial features. This unique design allows it to overcome the long-term dependency bottleneck of traditional encoders and effectively parse the coupling relationships between multi-dimensional features when dealing with complex AIS sequence features, making it the optimal choice for achieving deep feature extraction. Therefore, Transformer Encoder is selected as the feature extractor for the ETA prediction module in this study.

After fusing these two parts of features (i.e., average pooling), the Transformer Encoder further encodes the fused sequence features.The fully connected layer processes the features output by the Transformer encoder to obtain a time regression result in hours, which represents the estimated remaining time for the vessel to sail to the destination port. A Transformer Decoder is connected after the Transformer Encoder to decode the encoded feature vectors into tensors with the same size as the input tensors of the Transformer Encoder. In backpropagation, a loss is calculated and minimized between the decoded result and the input tensors of the Transformer Encoder, ensuring that the decoder's output remains as consistent as possible with the encoder's input. This ensures that the encoder can minimize the information loss caused during the encoding process.

In addition, turn around time at port (port stay time) is one of the key factors affecting ship ETA prediction. Therefore, this study introduces the turn around time at port from raw AIS sequence data as an important data to achieve more accurate ETA prediction. It should be clarified, however, that the predicted value is the sum of the turn around time and the model's output. The turn around time is not input into the model like the data in Table 1.

### 4.3 Loss Aggregation

In the backpropagation process, the initial loss magnitude of the port prediction module and feature reconstruction module is $10^0$, while that of the ETA prediction module is $10^3$. If the three losses are directly summed as the loss aggregation method, the model will predominantly focus on minimizing the loss of the ETA prediction module while neglecting that of the port prediction module and feature reconstruction module. Therefore, we multiply the loss of the ETA prediction module by a coefficient of 0.001 to make the three losses of the same magnitude.

In this multi-task prediction model, due to the difference in learning complexity between the port prediction module and the ETA prediction module, the fitting process of the former is earlier than that of the latter. In response to this characteristic, the model introduces a dynamic weight allocation mechanism: when it is detected that the port prediction module has completed fitting (i.e., no loss reduction for 10 epochs), the parameter update strategy in the backpropagation stage is dynamically adjusted from the initial average weight allocation to a higher weight for the ETA prediction module. This strategy aims to enhance the learning intensity of the ETA prediction module, promote its accelerated convergence in subsequent training, thereby achieving collaborative optimization among multiple tasks and improving the overall performance and generalization ability of the model.

$$L_{\text{total}} = \alpha \cdot L_{\text{port}} + 0.001(0.66 - \alpha) \cdot L_{\text{ETA}} + 0.34 \cdot L_{\text{feature reconstruction}} \qquad (1)$$

where $L_{\text{port}}$ is the loss of port prediction module, $L_{\text{ETA}}$ is the loss of the ETA prediction module, $L_{\text{feature reconstruction}}$ is the loss of the feature reconstruction module and $0 < \alpha \leqslant 0.33$; the initial value of alpha is 0.33, and it gradually decreases after the port prediction module is fitted.

## 5 Experiment

### 5.1 Dataset

The AIS trajectory data constructed in this study focus on the ETA prediction task of ships on intercontinental routes, covering two types of vessels: Crude oil tanks and container ships. Data

Table 3: Benchmark

| Model | Ship type | Spatial scopes | ETA Prediction | |
| --- | --- | --- | --- | --- |
| | | | MAE | MSE |
| El Mekkaoui et al. (2023) | Bulk ships | About 2500 nautical mile | 4.00 | 70.15 |
| Ogura et al. (2021) | N.A. | Japan-Taiwan & Taiwan-U.S.A. | 4.60 | - |
| **TransES-ETA** | Crude Oil Ship and Container Ship | About 2468 nautical mile | **2.74** **2.46** | **18.14** **14.51** |

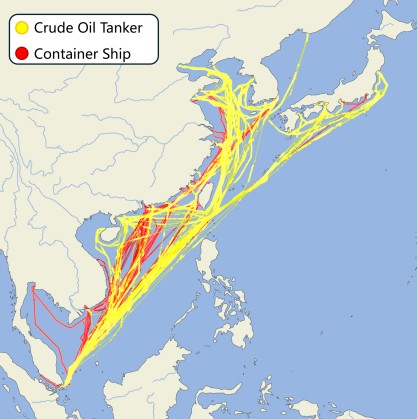

Figure 2: Visualization of Ship trajectory

collection areas cover seas such as the South China Sea, the East China Sea and the Yellow Sea, as shown in Fig. 2. As shown in Table 2, the dataset includes real navigation trajectories from 34 departure ports in countries such as China, South Korea, and Japan to the port of Singapore in 2024. Among them, the average navigation distance of a single trajectory reaches 2467.94 nautical miles (4570.62 kilometers). Note that all trajectories include intermediate calling ports, which results in dynamic changes in the routes from the origin port to the destination port. In this study, sliding window sampling is adopted, where each sliding window consists of ten grids with a size of 6×6 nautical miles.

## 5.2 Experiment Settings

Our pipeline is composed of GCN, Bi-GRU, Transformer Encoder, and Transformer Decoder. An Adam optimizer is adopted with a learning rate set at 0.001. The model is trained for 100 epochs with a batch size of 128. For the port prediction module, the Cross-Entropy loss function is applied, while the Mean Squared Error (MSE) loss function is used for both the ETA prediction module and the feature reconstruction module. Experiments show that using MSE as the loss function achieves better convergence results compared to Mean Absolute Error (MAE). The data set is divided into a training set and a test set in a ratio of 90% and 10%, respectively.

The experiments are conducted under the following environment: the GPU is NVIDIA RTX 4070, the CPU is Intel i7-14650HX, the memory size is 16 GB. The model is implemented using PyTorch 2.5.1 with Python 3.12.

## 5.3 Evaluation

This study employs accuracy, MSE and MAE as the metrics. Accuracy evaluates the proportion of correctly predicted samples by the model to the total number of samples, and it reflects the overall port prediction accuracy of the model. MSE is an indicator that measures the average squared error between the predicted values and the true values of a regression model, and it mainly evaluates the degree of dispersion of the ETA prediction model's errors. MAE is an indicator that measures the average absolute deviation between the model's predicted values and the true values, and it mainly

Table 4: GRU vs GCN

| Models | ETA prediction | | Time | |
|---|---|---|---|---|
| | MAE | MSE | Training | Testing |
| Original model | 3.32 | 30.83 | 215.00 minutes | 0.0181 seconds |
| Improved model | **2.46** | **14.52** | **41.00 minutes** | **0.0165 seconds** |

evaluates the overall average deviation level of the ETA prediction model's predictions. The detailed calculation are shown in supplementary materials.

We compare benchmark ETA models from prior studies with the deep learning architecture proposed here (as shown in Table 3, - indicates that this metric was not used in the study.). The port prediction module not only models the intermediate calling ports but also implicitly captures the corresponding port stay times, which are inherently intertwined with the characteristics of each port. To empirically validate this implicit modeling ability, we conducted comparative experiments. In the first experimental setup, port stay time were explicitly incorporated as additional input features to the model, while in the second setup, the model relied solely on its internal mechanisms to infer these durations without any explicit time-related input.

According to Table 3, the model proposed in this paper outperforms other studies in the same field, whether in port prediction or ETA prediction. It is worth noting that with the number of input elements being one-third of that in the study El Mekkaoui et al. (2023), the model MAE and MSE has decreased by 31.50% and 74.14% respectively, which confirms the superiority of this study. In addition, the MAE and MSE of the model without explicitly adding port stay time are only 0.29 and 3.63 higher than those of the model with explicitly added port stay time, respectively. This demonstrates that the port prediction module can effectively extract port stay features from AIS sequence information.

### 5.4 ABLATION STUDY

#### 5.4.1 GRU VS GCN

To verify the actual improvement effect of the novel AIS sequence data encoding method and GC-NwRC module proposed in this study on model performance, an ablation experiment is designed. By comparing the performance of the model before and after the improvement, it quantitatively analyzes the effectiveness and necessity of the module.

The original AIS sequence feature extraction module uses a GRU to extract **local patterns**. On the dataset of this study, the ETA prediction MAE and MSE are 3.32 hours and 30.83 hours², respectively, with a total training time of 215.00 minutes and a single-sample inference time of 0.0181 seconds, as shown in Table 4. By contrast, the improved model encodes sub-AIS sequence data into a directed complete graph and uses GCNwRC to extract features. For this study, it optimizes the problem that the original model cannot perform batch processing, and shortens both the model training time and inference time. The results show that under the same task, dataset and experimental settings, the MAE and MSE of ETA prediction are reduced to 2.45 and 14.52, down by 26.04% and 52.90% respectively; the total training time and single-sample inference time are reduced to 41.00 minutes and 0.0165 seconds, decreasing by 80.93% and 8.84% respectively, as shown in Table 4.

The experimental results indicate that improving this module can significantly reduce the model's training time. This optimization directly reduces the current model's training time costs and provides novel ideas for feature extraction methods of AIS sequence data, enabling R&D personnel to explore innovative directions at a faster pace. The dataset size of this experiment is small, and the importance of this module will be more prominent when experiments are conducted on larger-scale datasets in the future. From a macro perspective, this optimization is effective.

#### 5.4.2 PORT CALL PREDICTION MODULE FOR ETA PREDICTION

To verify the necessity of port prediction schedule module in the ETA prediction task, this ablation experiment is designed. By comparing the performance of the improved model with port call schedule module and the baseline model without port call schedule in the ETA prediction task, the

Table 5: Port Call schedule for ETA Prediction

| Models | ETA prediction | |
| --- | --- | --- |
| | MAE | MSE |
| Baseline model | 5.39 | 74.38 |
| Improved model | **2.46** | **14.51** |

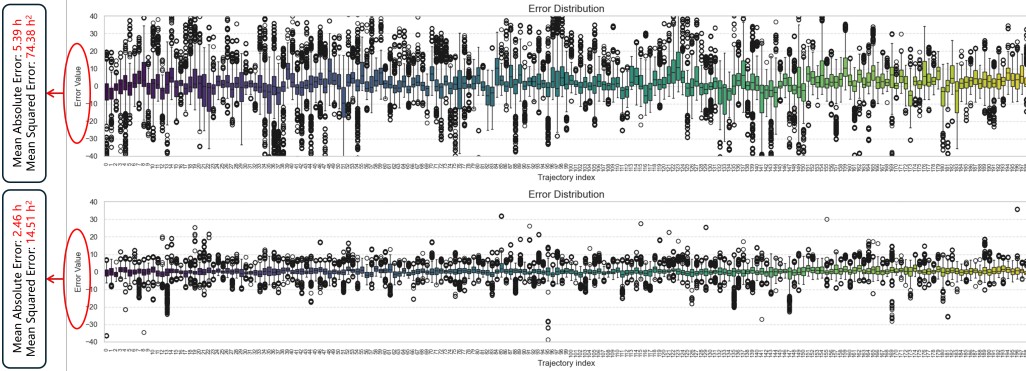

Figure 3: Box polt of error distribution

actual improvement effect of port call schedule module on the model is analyzed. A baseline model containing the AIS sequence data encoding module with GCNwRC and the Transformer Encoder-Decoder is constructed. It is trained and tested on the dataset of this paper, with MAE and MSE being 5.39 hours and 74.38 hours² respectively, as shown in Table 5. Subsequently, the port scheduling module is integrated into the above baseline model to form a complete improved model, and the experiment under the same conditions is conducted again. As shown in Table 5, the MAE and MSE of the model are optimized to 2.45 and 14.51, which are 54.55% and 80.49% lower than those of the baseline model. The experimental results fully demonstrate that the improved model not only enhances the average prediction accuracy of the overall samples (with a decrease in MAE) but also specifically suppresses the extreme deviations of a few samples (with a significant decrease in MSE), as shown in Fig. 3. This indicates that the port call schedule module can effectively strengthen multi-dimensional feature interaction, confirming its importance and necessity in improving the accuracy and robustness of ETA prediction.

## 6 CONCLUSION

ETA prediction are key topics in maritime AI, but traditional two-phase ML/DL methods overlook their interdependence, causing error accumulation and inefficiency. End-to-end models improve efficiency but lack interpretability, as they bypass intermediate port call schedules, reducing transparency. Thus, we proposed a TransES-ETA model to predict the port call schedule and final ETA of each vessel. This end-to-end pipeline is designed to predict two tasks simultaneously, with the losses from each task aggregated during training to learn a shared, robust data representation. This joint optimization not only improves the performance of both tasks but also makes the overall prediction process more efficient for both training and inference. Since port scheduling and ETA prediction are inherently interrelated, incorporating port call schedule features into the ETA prediction process makes the results clearer and more interpretable, as port call schedules serve as essential inputs that guide and contextualize the final ETA estimation. Moreover, our data processing strategy and the design of our transformer-based model incorporate novel modules specifically crafted to enhance both predictive accuracy and computational efficiency, ensuring superior final results.

To the long run, this work introduces and validates a novel approach for addressing highly inter-related tasks through a unified pipeline in the maritime AI domain. The proposed framework can be extended to other related tasks in this field, offering potential improvements in both predictive accuracy and computational efficiency.

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

## A  APPENDIX

### A.1  SPATIAL ENCODING

$$\text{SE}_{(\lambda,\phi,4i)} = \cos(g(\phi,4i)) \cdot \sin(g(\lambda,4i))$$
$$\text{SE}_{(\lambda,\phi,4i+1)} = (\log \pi)^2 \cdot \sin(g(\phi,4i))$$
$$\text{SE}_{(\lambda,\phi,4i+2)} = \cos(g(\phi,4i)) \cdot \cos(g(\lambda,4i)) \tag{2}$$
$$\text{SE}_{(\lambda,\phi,4i+3)} = -(\log \pi)^2 \cdot \sin(g(\phi,4i))$$

where the function $g(q,4i) = q/(2\pi)^{4i/d^2}$ operates with radian unit $(\lambda,\phi)$, and $i$ specifies the dimension. First, each component combination is based on the transformation logic from polar coordinates to Cartesian coordinates, and different dimensions ($4i$, $4i+2$, $4i+1$, $4i+3$) serve to represent different wavelength ranges. Specifically, the $4i$ and $4i+2$ dimensions are based on wavelengths ranging from $\pi$ to $\pi \cdot (2\pi)^{-1/d}$, while the $4i+1$ and $4i+3$ dimensions correspond to wavelengths ranging from $2\pi$ to $(2\pi)^{1-1/d}$. Second, the function $g(q,4i) = q/(2\pi)^{4i/d^2}$ operates on radian units $(\lambda,\varphi)$, where $i$ specifies the dimension. This definition is standardized by considering the polar-to-Cartesian coordinate transformation context and the explanation that the frequency is determined by the $2\pi$ term in the denominator of $g(q,4i)$. Finally, $(\log \pi)^2$ serves as a scaling factor to match the polar-to-Cartesian coordinate transformation and allows the output of the SE module to preserve the spatial distance concept of the original coordinates, enabling the encoding vector to represent positions.

### A.2  TIME ENCODING

By modifying the position encoding formula of Transformer Vaswani et al. (2023), we obtain the time encoding formula for *Time Distance*:

$$\text{TE}_{(\Delta,2i)} = \cos\left(\Delta/1000^{2i/d}\right)$$
$$\text{TE}_{(\Delta,2i+1)} = \sin\left(\Delta/1000^{2i/d}\right) \tag{3}$$

where $i$ specifies the dimension; $\Delta$ is time distance; $d$ is feature dimension. To ensure sufficient coverage of the maximum temporal progression of vessel operations while capturing adequate intervals in the encoded representations, the frequency of the sinusoidal functions in TE is set to 1000.

## B  STATEMENT

### B.1  LLM USAGE FOR PAPER WRITING

We acknowledge the use of large language models (LLMs) to support and refine our writing process. Specifically, we drafted the initial version of the manuscript ourselves and then utilized ChatGPT-4o to assist with grammar correction and stylistic polishing in accordance with academic writing standards. All generated suggestions were carefully reviewed, and any changes were made with the intent of preserving the original meaning and scientific accuracy. We take full responsibility for the final content of this paper.

