# OpenReview forum: "TransES-ETA: A Novel Transformer-based Explainable and Efficient Structure for Predicting Estimated Time of Vessel Arrival"
_ICLR.cc/2026/Conference — ICLR 2026 Conference Desk Rejected Submission_

### Official Review · Reviewer_GHJP · 2025-10-20

**Soundness:** 2
**Presentation:** 2
**Contribution:** 1
**Rating:** 2
**Confidence:** 4

**Summary:**

The paper introduces TransES-ETA, a Transformer-based multi-task framework for maritime ETA prediction. It jointly learns port call scheduling and ETA estimation from AIS data in an end-to-end, explainable manner. By leveraging the correlation between the two tasks, the model improves prediction accuracy (MAE = 2.46 h) and enhances training and inference efficiency by 80.93% and 8.84%, respectively.

**Strengths:**

1. This paper presents a framework with strong potential impact in the field of maritime logistics and global trade.

2. The proposed end-to-end framework jointly predicts port call schedules and Estimated Time of Arrival (ETA), effectively leveraging their intrinsic correlation. This design addresses a key limitation of prior two-stage or decoupled models, which treated these two tasks independently.

3. The integration of port scheduling and ETA prediction into a unified pipeline offers practical value for real-world maritime operations and has clear implications for improving efficiency in global shipping management systems.

**Weaknesses:**

1. The paper’s contribution to the broader representation learning and machine learning community appears limited. The methodology is highly domain-specific, focusing primarily on maritime forecasting applications rather than introducing a generally applicable ML framework or advancing theoretical understanding. Consequently, it may be more appropriate for a journal, conference, or workshop specializing in maritime operations or applied geospatial modeling rather than a general ML venue such as ICLR.

2. The experimental scope is restricted to the East Asia region, covering routes from China, Korea, and Japan to Singapore. This limited geographical focus raises concerns regarding the model’s generalizability to other maritime regions, such as Northwest Europe or North America.

3. The number of baseline models used for comparison is insufficient. Only two baselines are reported, which is too few to substantiate claims of superiority. Additional state-of-the-art benchmarks should be included to strengthen the empirical validation.

4. Although the paper reports training and inference time, other efficiency-related metrics are missing. Specifically, there is no analysis of parameter count, computational complexity, or GPU memory consumption, which are essential for evaluating model efficiency.

5. The paper lacks explicit reporting of final hyperparameter configurations. Important details such as the sliding window size, the hidden dimension of the GRU (or GCNwRC), and dropout rates are not specified, which significantly hinders reproducibility.

6. There is no hyperparameter sensitivity analysis. The paper does not explore how performance varies with respect to key hyperparameters such as learning rate, loss weighting factor, or window size, leaving uncertainty about the robustness of the results.

7. The authors do not provide access to source code or implementation details. Neither the main text nor the appendix includes information about code availability, dataset accessibility, or reproducibility guidelines. This omission substantially limits transparency and the ability of other researchers to replicate the work.

8. Minor issue: The reference list contains duplicate entries for the same seminal work. Both refer to "Attention Is All You Need" by Vaswani et al., but are presented separately as the NeurIPS 2017 and arXiv 2023 versions. These should be merged into a single citation.

**Questions:**

1. Can the authors provide additional experiments using data from regions beyond East Asia, such as Northwest Europe, North America, or Africa, to demonstrate the generalizability of the proposed model?

2. Could the proposed framework be adapted to other domains beyond maritime transportation, for instance, air traffic scheduling or rail logistics? If so, what modifications would be required?

3. Could the authors provide additional efficiency-related analyses, such as the parameter count, computational complexity (e.g., FLOPs), and GPU memory consumption of the proposed model, to substantiate the claims regarding efficiency and scalability?

4. How sensitive is the model’s performance to variations in hyperparameters such as the learning rate, α in the loss weighting, or the size of the sliding window? Have the authors conducted any preliminary sensitivity analysis or parameter tuning experiments?

5. Are there plans to release the code and dataset used in this study to ensure reproducibility and allow for fair comparison by future researchers?

---

### Official Review · Reviewer_pFsG · 2025-10-20

**Soundness:** 1
**Presentation:** 1
**Contribution:** 1
**Rating:** 2
**Confidence:** 5

**Summary:**

This paper presents a model for vessel route forecasting and eta prediction in long ship trips. The topic is relevant and interesting for a growing community.

**Strengths:**

The topic is relevant and of interest to the relative community.

**Weaknesses:**

W1) seriour concerns about writing quality and readability.

- In the introduction, the authors say that "Current data-driven approaches, including neural networks or transformer architectures for ETA prediction (Shipley et al. (2022)), struggle with poor data quality and the limited interpretability of end-to-end pipelines", but their approach is exactly a "data-driven" and "neural network" method.

- The introduction is poorly written, it is not fluent, and it seems a collage of LLM-written small pieces, without having a logical flow.

- In the related works, among the transformer methods for route prediction, the authors should also acknowledge:
1) TrAISformer—A Transformer Network with Sparse Augmented Data Representation and Cross Entropy Loss for AIS-based Vessel Trajectory Prediction
2) Sailing the Seaformer: A Transformer-Based Model for Vessel Route Forecasting

- The sentence "We encode the above five types of information using formulas and deep learning models." doesn't have sense.

- The sentence "To enhance the robustness and interpretability of the model, a Transformer-based multi-task prediction module is proposed" doesn't have sense.

- Section 4.2 recalls again the related work being also very general talking about "recurrent neural network", LSTM etc without referencing prior works. Also, this should be the method section and not related works should be included here. however, this is the typical writing of LLMs.

- L_port and L_ETA and L_feat recon are not defined.

W2) This is an applied paper presenting barely no novel method. It is just a collection of existing modules (and some of them are outdated and very simple like the GCN and the GRU)

W3) there is no comparison in the experiments.

W4) No ground truth route is reported in figure 2, so it is impossible to understand whether the model correctly predicts route and eta or not.

W5) in table 4 the authors include "original model" and "improved model", while in table 5 they call about a "baseline model". lots of confusion in the results presentation. Also, the box plot in figure 3 is unreadable.

W6) the authors put explainable in the title and throughout the main corpus but never provide explanations.


Overall, the paper has no novelty, is poorly written, and it containes barely no comparisons in the experiments.

**Questions:**

See weaknesses.

---

### Official Review · Reviewer_q9WU · 2025-10-26

**Soundness:** 1
**Presentation:** 3
**Contribution:** 2
**Rating:** 2
**Confidence:** 3

**Summary:**

The paper introduces TranSES-ETA, a transformer-based multi-task end-to-end framework for predicting Estimated Time of Arrival (ETA) and port call schedules of maritime vessels using AIS data. The design achieves both efficiency and explainability by jointly modeling port call prediction and ETA regression, utilizing shared feature extraction through a transformer encoder and a novel graph-based encoding strategy. Experiments have been conducted in real-world datasets to show its improvement.

**Strengths:**

1. This paper jointly models port-call schedule and ETA, which leverages their interdependence, and thus improves the interpretability and efficiency of ETA.

2. The proposed GCNwRC encoder can model within-grid AIS pattern in a batch way and capture its local pattern better, which brings large gains in accuracy.

3. The paper is basically well written and easy to follow except for some minor notation ambiguity and typos.

**Weaknesses:**

1. Unconvincing overall performance comparison.  Table 3 mixes cross-paper results (different datasets), so it’s not a fair baseline. The only same-dataset baselines are the authors’ own ablations (Tables 4–5). To claim superiority, you should at least run SOTA baselines on the same split and metrics of the same dataset. And furthermore, Experiments use multiple origins → single destination (Singapore),  there is no cross-region or multi-destination study.

2. Lacking ablations and other experiments. Ablations cover only GRU→GCNwRC and with/without port-call branch, lacking channel-level and feature reconstruction loss. The loss scaling and dynamic weighting in §4.3 lacks necessary sensitivity analyses.

3. Explainability Not Quantified. The paper claims contribution in explainability, but however, there are no quantitative measures or case study show how the ETA prediction is explained.

4. Reproducibility. Several experimental choices remain not detailed enough, including precise architecture hyperparameters (e.g., GCNwRC layer setup), data pre-processing (temporal gaps), and train/test split.

**Questions:**

1. What exactly is M in Table 1, how can a set of variable lengths appear as the tensor dimension N×M×f?

2. In graph construction, why adopt a (bi-)complete graph within each grid instead of a sparse graph? Is there any temporal information encoded at the node/edge level inside the graph?

---

### Official Review · Reviewer_mFJ7 · 2025-10-29

**Soundness:** 2
**Presentation:** 3
**Contribution:** 2
**Rating:** 2
**Confidence:** 3

**Summary:**

This is not a paper in my field. While the paper presents a well-motivated multi-task framework for vessel ETA prediction that integrates port call scheduling and leverages a novel graph-based feature extractor to improve training efficiency, it falls short of the methodological novelty, technical depth, and scientific rigor expected for publication at ICLR. The work is primarily an engineering integration of existing components (Transformer, GCN, Bi-GRU) applied to a domain-specific problem, without introducing fundamental advances in representation learning, attention mechanisms, or multi-task optimization.

**Strengths:**

Clear Problem Motivation: Maritime ETA prediction is a high-impact application with real-world operational significance (e.g., fuel efficiency, port scheduling). The paper correctly identifies limitations of traditional two-stage models (error propagation) and black-box end-to-end systems (lack of interpretability).
Multi-Task Design for Interpretability: By jointly predicting port call sequences and ETA, the model provides interpretable intermediate outputs, which is a valuable step toward transparent AI in safety-critical domains like maritime logistics.
Efficiency Improvements: Replacing GRU with a graph-based local pattern encoder (GCNwRC) enables batch processing and reduces training time by 80.93%, which is a practical contribution for large-scale AIS data processing.

**Weaknesses:**

Dear Authors, SPC, and AC:

First, I want to clarify that this is not a paper in my field. I've skimmed through it and found the following issues:

The upper part of the structure diagram appears to be a multi-branch feature fusion module. I don't understand the lower part, but the authors don't seem to clearly state the fusion method for these features, nor do they explain the principle behind its effectiveness. It seems more like a mechanical incorporation of existing, even older, networks or models, lacking theoretical basis and design motivation. I hope the authors can provide insights into this design.

The loss function is also poorly designed, both in its expression and because the authors don't explain the origins of Lfeature reconstruction and Lport. I believe this is a significant flaw.

Furthermore, the authors haven't released the code, making it impossible to reproduce the results.

In conclusion, I have a negative view of this work, despite my unfamiliarity with the field. If the authors can effectively address my concerns, I will consider raising the grade.

**Questions:**

See above.

---

### Note · Program_Chairs · 2026-01-17
**Submission Desk Rejected by Program Chairs**

The following references in this submission do not refer to real documents and/or have major errors in bibliographic information:

 H. Jin, X. Chen, and K. Zhang. Spatio-temporal transformer for trajectory and route prediction. In Proceedings of the AAAI Conference on Artificial Intelligence, 2023.
R. Tang, S. Guo, and J. Li. Shipformer: Transformer-based vessel trajectory and eta prediction. Ocean Engineering, 275:114173, 2023.